



# An empirical zenith wet delay correction model using
# piecewise height functions
YiBin Yao[1,2,3], YuFeng Hu[1*]
[1]*School of Geodesy and Geomatics, Wuhan University,* 129 *Luoyu Road, Wuhan* 430079*, China.*
[2]*Key Laboratory of Geospace Environment and Geodesy, Ministry of Education, Wuhan University,*
129 *Luoyu Road, Wuhan* 430079*, China*
[3]*Collaborative Innovation Center for Geospatial Technology,* 129 *Luoyu Road, Wuhan* 430079*,*
*China*
*Corresponding to: yfhu@whu.edu.cn*
***Abstract**—*Tropospheric delay is an important error source in space geodetic techniques.
The temporal and spatial variations of the zenith wet delay (ZWD) are very large, and
thus limit the accuracy of tropospheric delay modelling. Thus it is worthwhile
undertaking research aimed at constructing a precise ZWD model. Traditional
tropospheric modelling methods do not consider the effects of different heights on
ZWD. Based on the analysis of vertical variations of ZWD, we divided the troposphere
into three height intervals: below 2 km, 2 km to 5 km, and 5 km to 10 km, and
determined the fitting functions for the ZWD within these height intervals. The global
ZWD model HZWD, which considers the periodic variations of ZWD with a spatial
resolution of $5\degree \times 5\degree$, is established using the ECMWF ZWD profiles from 2001 to
2010. Validated by the ECMWF ZWD data in 2015, the precisions of the ZWD
estimation in the HZWD model over the three height intervals are improved by 1.4 mm,
0.9 mm, and 1.2 mm, respectively, compared to that of the currently best GPT2w model
(23.8 mm, 13.1 mm, and 2.6 mm). The test results from ZWD data from 318 radiosonde
stations show that the root mean square (RMS) error in the HZWD model over the three
height intervals was reduced by 2%, 5%, and 33%, respectively, compared to the
GPT2w model (30.1 mm, 15.8 mm, and 3.5 mm) over the three height intervals. In
addition, the spatial and temporal stabilities of the HZWD model are higher than those
of GPT2w and UNB3m.




***Index Terms***—Tropospheric delay, zenith wet delay, vertical variations, height dividing,
HZWD model.

## 1 Introduction

The radio waves experience propagation delays when passing through the neutral

atmosphere (primarily the troposphere), which are known as the tropospheric delays.
The tropospheric delay is one of the main error source in space geodetic techniques. In
the processing of the space geodetic data, the tropospheric delay along the propagation
path is generally expressed as the product of zenith tropospheric delay (ZTD) and
mapping function (MF). The ZTD is divided into a zenith hydrostatic delay (ZHD) and
a zenith wet delay (ZWD) (Davis *et al*., 1985), and the ZHD can be accurately
determined using pressure observations. Unlike the ZHD, the ZWD is difficult to
calculate accurately due to the high spatio-temporal variation in water vapour, thus
making itself the main factor influencing tropospheric delay correction.

The traditional Saastamoinen model (1972) and Hopfield model (1971)

approximate the ZWD with temperature and water vapour pressure observations.
Without considering the vertical distribution of water vapour, the stability and reliability
of their ZWD estimates are poor. Moreover, both models are highly dependent on
meteorological data, which greatly limits their application in wide area augmentation
and real-time navigation and positioning. Therefore, non-meteorological parameters-
based models were proposed as practical conditions required. The RTCA-MOPS (2016),
designed by the US Wide Area Augmentation System (Collins *et al*., 1996), estimates
ZWD by using the latitude band parameters table. The modified RTCA-MOPS model
– called UNB3m (Leandro, 2006) – uses relative humidity as a parameter instead of the
water vapour pressure to calculate the ZWD, effectively improving the precision of
ZWD estimation compared with previous model versions, but the model deviation is
increased when the height exceeds 2 km. The TropGrid model (Krueger *et al.*, 2004,
2005) provides the meteorological parameters needed to calculate tropospheric delay in





the form of $1°\times1°$ grid. The TropGrid model calculates ZWD with modelled water
vapour pressure and weighted mean temperature data, while the TropGrid2 model
(Schüler, 2014) directly models ZWD and uses the exponential function to describe the
variation of ZWD with respect to height, resulting in a precision improvement. Based
on the GPT2 model (Lagler *et al*., 2013), the GPT2w model (Böhm *et al*., 2015) adds
weighted mean temperature and a vapour pressure decrease factor realised as a global
grid to estimate ZWD by using the Askne and Nordius formula (Askne & Nordius,
1987). The GPT2w model has the best performance with regard to ZWD estimation
compared to other commonly used models (Möller *et al*., 2014).
The water vapour changes rapidly with respect to height, and the trends in water
vapour at different heights vary, so the wet delay with direct relation to water vapour
has complex spatio-temporal variations in the vertical direction. The aforementioned
troposphere models are all based on a fixed height (average sea level or surface height)
and use only a single decrease factor to describe the variation of water vapour or wet
delay with respect to height, which makes it difficult to allow for the vertical
distribution differences in water vapour (or wet delay) in the upper troposphere. In the
course of aircraft dynamic navigation and positioning, it is necessary to correct the wet
delay at different heights, which is obviously difficult for the aforementioned models.
Based on the analysis of the characteristics of the ZWD profile, an empirical ZWD
model, named HZWD, is established based on three functions applicable within
corresponding height intervals, and the model precision is verified by European Centre
for Medium-Range Weather Forecast (ECMWF) reanalysis data as well as radiosonde
data.

**2 Vertical variations of ZWD**
ZWD is defined as the integral of the wet refractivity along the vertical profile
above the station:
$$ZWD = 10^{-6} \int_{H}^{\infty} N_w dh = 10^{-6} \int_{H}^{\infty} (k_2' \frac{e}{T} + k_3 \frac{e}{T^2}) dh \qquad (1)$$





In equation (1), $N_w$ is the wet refractivity; $e$ is the water vapour pressure in hPa;
$T$ is the temperature in Kelvin; $k_2'$ is 17 K/hPa and $k_3$ is 377600 K$^2$/hPa (Bevis *et al.*,
1992). It can be seen from equation (1) that ZWD changes with height, vapour pressure
and temperature. The ZWD will decrease with increasing height due to the shorter
integral length. The accurate ZWD calculation requires profiles of water vapour
pressure and temperature, which are difficult to access in practical applications (such
as aircraft navigation and positioning and wide area augmentation). Therefore, it is
necessary to develop an empirical ZWD model with high precision. The temperature
roughly decreases linearly with increasing height in the troposphere, while the change
in water vapour is more variable, so the water vapour is the main determinant of vertical
variation of ZWD. In the following content, we used the meteorological data profile of
ERA-Interim pressure levels provided by ECMWF to analyse the vertical variation
characteristics of ZWD and explore a suitable fitting function capable of describing the
changes in ZWD with respect to height.
ERA-Interim can provide data at 0:00, 6:00, 12:00, and 18:00 UTC daily with a
spatial resolution of not more than $0.125\,°\times0.125\,°$ and 37 pressure levels. The highest
level data come from a height of approximately 50 km, covering almost the entire
troposphere and stratosphere. We used the temperature, the geopotential height, and the
specific humidity provided by the ERA-Interim pressure levels data, and the discretised
form of equation (1), to calculate the ZWD for each level height:
$$\begin{cases} e_i = q_i \times P_i / (0.622 + 0.378 \times q_i) \\ N_{w_i} = k_2' \dfrac{e_i}{T_i} + k_3 \dfrac{e_i}{T_i^2} \\ ZWD = 10^{-6} \sum_i^{36} \dfrac{N_{w_i} + N_{w_{i+1}}}{2} \cdot \left(h_{i+1} - h_i\right) \end{cases}$$
(2)

In equation (2), $e$ is the water vapour pressure in hPa; $q$ is the specific humidity in g/g;
$P$ is the pressure in hPa; $T$ is the temperature in kelvin; $k_2'$ and $k_3$ are empirical
constants same as equation (1); $h$ is the geopotential height in meters. From equation





(2), we can see that the ZWD at specific level height is the sum of the ZWD portions in
all layers above the specific level height. Figure 1 shows the water vapour pressure and
ZWD profiles at a grid point (0 °N, 0 °E) at 12:00 UTC on 1 January, 2010. From Figure
1, it can be seen that the downward trend in the water vapour pressure varies
significantly with height, and the decrease factor is different across different height
intervals. The changes in ZWD with respect to height are similar to that of the water
vapour pressure with respect to height: the decay is fastest up to a few kilometres height
and slows down with increasing height; the ZWD values are close to zero after 10 km.
Zhao *et al*. (2014) showed that about 50% of the water vapour content is concentrated
within 1.5 km of the surface and less than 10% of the water vapour content remains
above 5 km, leading to different ZWD decay rates within different height intervals.
These results are basically consistent with our experiment results. Figure 2a shows the
variation of ZWD vertical gradients with respect to height. From Figure 2a, it can be
seen that the trends in ZWD vertical gradients at different height intervals are obviously
different. Specifically, the linear fit of the ZWD gradients with height below 2 km
shows a great agreement with an R square value of 0.99 (Figure 2b). Thus we can come
to a conclusion: ZWD gradients roughly change linearly below 2 km; and from 2 km to
5 km, and 5 km to 10 km, the ZWD gradients vary non-linearly.

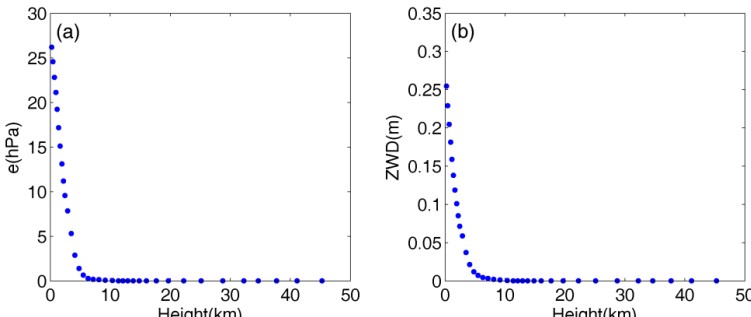


Figure 1 Water vapour pressure (a) and ZWD (b) *versus* height.






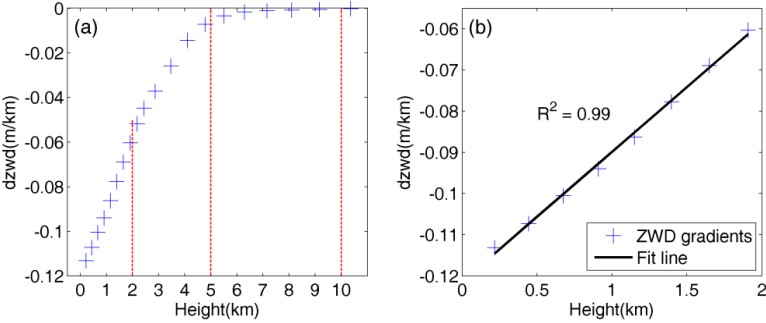


Figure 2 ZWD vertical gradients with respect to height (a) and linear fit with height below 2 km
(b).
Figure 3 shows the ZWD vertical gradients with respect to height at grid points in
different latitude bands. Figure 4 shows the similar ZWD vertical gradients as Figure 3
but for different season. The variations are similar to those in Figure 2a, which show
trend changes at about 2 km and 5 km. It is worth noting that the ZWD gradients at high
latitudes are much larger and water vapour is more variable than at low latitudes,
resulting from the fact that the water vapour at high latitudes are more variable. In
addition, the ZWD gradient trends in the southern hemisphere are significant. In
contrast, the ZWD gradients in the northern hemisphere are slightly complicated with
respect to height: the reason for this may be that the southern hemisphere is mostly
oceanic while the northern hemisphere has many seacoasts. The terrain complexity in
the northern hemisphere contributes to the disturbances in the ZWD gradient in specific
areas. According to the vertical variation characteristics of ZWD, we divided the
troposphere into three height intervals: below 2 km, 2 km to 5 km, and 5 km to 10 km,
and assumed 10 km as the empirical tropopause beyond which the ZWD is assumed to
be zero. For ZWD fitting with respect to height, TropGrid2 and GPT2w use exponential
functions, while some scholars have also used a polynomial to describe the tropospheric
delay with respect to height (Song *et al*., 2011). We used both polynomial and
exponential functions to fit the variation trend of the ZWD with respect to height in the
three selected intervals, respectively. The results showed that the quadratic polynomial
used under 2 km, and exponential functions between 2 km and 5 km, and 5 km to 10




km gave the best fits. The combination of the quadratic polynomial and exponential
functions for different height intervals is termed piecewise height functions. Table 1
summarises the global fitting statistics of different fit functions, demonstrating the
superiority of piecewise height functions to the single polynomial function and single
exponential function.

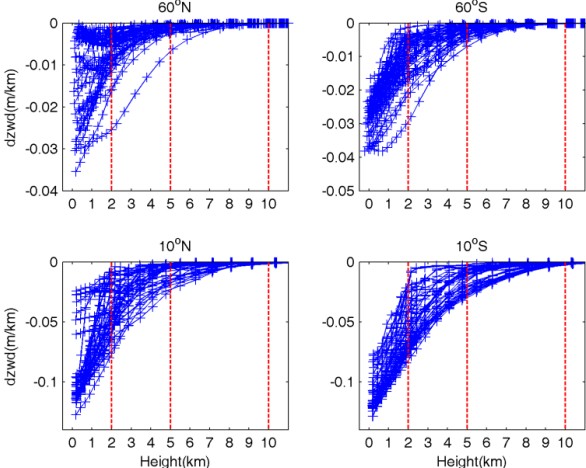


Figure 3 ZWD gradients with respect to height at grid points in different latitude bands (12:00
UTC, 1 January, 2010).

Figure 4 ZWD gradients with respect to height at grid points in different latitude bands (12:00
UTC, 1 July, 2010).



Table 1. Fitting RMSs of piecewise height functions, single quadratic polynomial function, and
single exponential function (unit: mm).

|  | < 2 km | 2 km to 5 km | 5 km to 10 km |
| --- | --- | --- | --- |
| Piecewise height functions | 0.2 | 1.0 | 0.2 |
| Quadratic polynomial | 5.9 | 3.8 | 6.5 |
| Exponential | 2.3 | 2.2 | 1.0 |


## 3 The HZWD model

From the above analysis of ZWD vertical variation and fitting, the piecewise height
functions of the proposed HZWD model are:
$$ZWD(B,L,H) = \begin{cases} z_1 + a_1 \cdot h + a_2 \cdot h^2 & h < 2000\ m \\ z_2 \cdot \exp\{\beta_2 \cdot (h - 2000)\} & 2000\ m \leq h < 5000\ m \\ z_3 \cdot \exp\{\beta_3 \cdot (h - 5000)\} & 5000\ m \leq h \leq 10000\ m \\ 0 & h > 10000\ m \end{cases} \tag{3}$$

In equation (3), $B$ is the latitude in degrees; $L$ is the longitude in degrees; $H$ is the
height in meters; function coefficients $z_1, z_2$ and $z_3$ can be regarded as the ZWD at
the height of 0 km, 2 km and 5 km, respectively. We used the monthly mean profiles of
ERA-Interim pressure levels from 2001 to 2010 with a horizontal resolution of $5° \times 5°$
for ZWD modelling. The ZWD profiles calculated for each grid point are fitted by
equation (3) to obtain the time series of the corresponding function coefficients: $z_1$,
$a_1$, $a_2$, $z_2$, $\beta_2$, $z_3$, and $\beta_3$. Jin *et al.* (2007) found that the tropospheric delay has
notable seasonal variations, mainly on annual and semi-annual cycles. Song *et al.* (2011)
and Zhao *et al.* (2014) considered the temporal features of function coefficients in their
troposphere models. We used the ten-year time series of those coefficients obtained to
analyse their temporal variations. Figure 5 shows the time series and cycle fitting results
of the function coefficients $z_1$, $z_2$, and $z_3$ at grid point (0 °N, 0 °E). Figure 5 shows
that the time series of the function coefficients $z_1$, $z_2$, and $z_3$ have a significant
characteristic annual cycle, and the semi-annual cycle is small but nevertheless obvious.


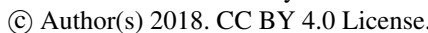



Figure 5 Time series and cycle fitting results of function coefficients $z_1$ (a), $z_2$ (b), and $z_3$ (c).

Therefore, taking the annual, and semi-annual, cycles into consideration, we used

equation (4) to fit the function coefficients derived from equation (3) to temporal
parameters for each grid point (Böhm *et al.*, 2015):

$$r(t) = A_0 + A_1 \cos\left(\frac{doy}{365.25} 2\pi\right) + B_1 \sin\left(\frac{doy}{365.25} 2\pi\right) \\ + A_2 \cos\left(\frac{doy}{365.25} 4\pi\right) + B_2 \sin\left(\frac{doy}{365.25} 4\pi\right)$$

(4)

In equation (4), $A_0$ is the annual mean; $A_1$ and $B_1$ are the annual cycle

parameters; $A_2$ and $B_2$ are the semi-annual cycle parameters; and *doy* is the day of
the year. It should be noted that the fitting results of coefficients $a_2$, $\beta_2$, and $\beta_3$
showed that all their annual means, and annual, and semi-annual, amplitudes are small.



However, below 2 km, the lack of cycle terms in $a_2$ would cause centimetre level error
in the ZWD estimates, so these terms have been retained. For $\beta_2$ and $\beta_3$, ZWD itself
is small at heights above 2 km, so the annual mean suffices for a desirable ZWD
estimate. The experiment revealed that the loss of accuracy due to the lack of annual
and semi-annual terms in $\beta_2$ and $\beta_3$ for the ZWD estimates is less than 0.1 mm.
Therefore, only the annual means are retained for these two coefficients.

Figure 6 shows the global distributions of annual means of model coefficients $z_1$,

$z_2$, and $z_3$. From Figure 6 we can see that the extremum of ZWD annual means at 0 m
height occur near the equator and the maximum exceeds 0.36 m. The ZWD annual
means decrease with increasing latitude. The distributions of ZWD annual means at 2
km and 5 km heights are similar to that at 0 m, but the areas with the large values near
the equator decrease in extent and the ZWD distributions tend to be uniform, indicating
that the water vapour content near the equator is greater than that in other regions, and
the ZWD value is also larger in low altitude regions. As the height increases, the
difference in water vapour content or ZWD, between the equator and other areas begins
to decrease, but remains significant. Overall, there are some differences in the ZWD
distribution at different heights, and it is necessary to model the spatio-temporal
variations of ZWD at different heights.





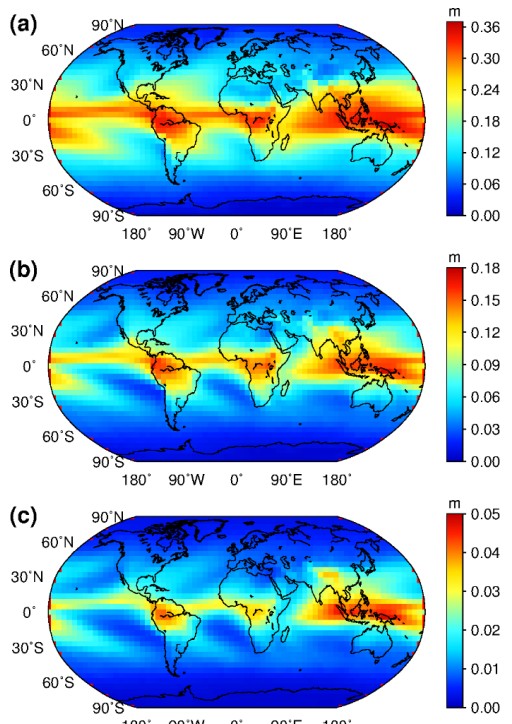


Figure 6 Global distributions of annual means of model coefficients $z_1$ (a), $z_2$ (b), and $z_3$

(c).

After the fitting processes involving equations (3) and (4), the global ZWD model

HZWD, using piecewise height functions, is established. The spatial resolution of the
HZWD model is $5° \times 5°$. For each grid point, there are 27 parameters which are stored
in text format. When the HZWD model is applied, the four grid points surrounding the
station are determined according to the horizontal position (latitude and longitude) of
the station, and then the model coefficients of the corresponding height intervals at the
four selected points are calculated according to equation (4). The ZWD of the four grid
points are extrapolated to the station height by using equation (3), and finally the ZWD
at the station location is obtained by using bilinear interpolation. The HZWD model
only needs time, latitude, longitude, and height as input parameters. It can calculate
ZWD without meteorological data, and can provide wet delay correction products for
navigation and positioning at different heights.





## 4 Validation and analysis of the HZWD model

To test the precision of HZWD model and analyse the model correction performance compared to other troposphere models, we used the ERA-Interim pressure levels data and radiosonde data from the year 2015 as external data sources, and compared the results with the commonly used models UNB3m and GPT2w. The parameters used for the validation are bias and RMS expressed as:

$$bias = \frac{1}{n}\sum_{i=1}^{n}(ZWD_i^M - ZWD_i^0) \tag{5}$$

$$RMS = \sqrt{\frac{1}{n}\sum_{i=1}^{n}(ZWD_i^M - ZWD_i^0)^2} \tag{6}$$

In equation (5) and (6), $ZWD_i^M$ is the value estimated by the model and $ZWD_i^0$ is the reference value.

For the UNB3m model, the ZWD at mean sea level (MSL) is first calculated, then a vertical correction is applied to transform the ZWD to the target height. The formulae are (Leandro *et al.*, 2006):

$$\begin{cases} ZWD_0 = 10^{-6}\dfrac{(T_m k_2' + k_3)R_d}{g_m(\lambda+1) - \beta R_d}\cdot\dfrac{e_0}{T_0} \\ ZWD = ZWD_0\left(1 - \dfrac{\beta H}{T_0}\right)^{\frac{(\lambda+1)g}{\beta R_d}-1} \end{cases} \tag{7}$$

where $T_m$ is the weighted mean temperature; $R_d$ is the specific gas constant for dry air; $g_m$ is the gravity acceleration at the mass centre of the vertical column of the atmosphere; $\beta$ and $\lambda$ are the temperature lapse rate and water vapour decrease factor, respectively.

For the GPT2w model, the modelled meteorological parameters at the four grid points surrounding the target location are extrapolated vertically to the desired height, then the Askne and Nordius formula (8) is used to calculate the wet delays at those base points: finally the wet delays are interpolated to the observation site in horizontal direction to get the target ZWD. It should be noted that the GPT2w model provides both $1°\times1°$ and $5°\times5°$ resolution versions. Since the horizontal resolution of HZWD model





is $5° \times 5°$, we used the GPT2w model with the same resolution for validation.

$$ZWD = 10^{-6} \cdot (k_2' + \frac{k_3}{T_m}) \cdot \frac{R_d e}{(\lambda+1)g_m} \qquad (8)$$


### 4.1 Validation with ECMWF data


Modelling of the HZWD model is based on the monthly mean profiles of ERA-
Interim pressure levels data from 2001 to 2010, while we used the ERA-Interim
pressure levels data with the full time resolution of 6 hours in 2015 for the model
validation. Regarding the ZWD profiles calculated from these data as reference values,
we calculated the global annual average bias and RMS error of the ZWD for three
models (HZWD, GPT2w, and UNB3m) within the three height intervals: below 2 km,
2 km to 5 km, and 5 km to 10 km (Table 2).
Table 2 Error statistics for the three models compared to the 2015 ECMWF data (unit: mm).

|  | < 2 km | | 2 km to 5 km | | 5 km to 10 km | |
|---|---|---|---|---|---|---|
|  | bias | RMS | bias | RMS | bias | RMS |
| HZWD | -2.0 | 23.8 | -1.4 | 13.1 | 0.0 | 2.6 |
| GPT2w | -0.1 | 25.2 | 2.5 | 14.0 | 2.2 | 3.8 |
| UNB3m | 16.6 | 41.4 | 10.9 | 22.7 | 3.5 | 5.8 |

From Table 2, it can be seen that the HZWD model is the most accurate model
across all three intervals, followed by the GPT2w model, and the UNB3m model has
the worst performance. The annual average biases of the HZWD model are lower than
that of the GPT2w model and the UNB3m model except below 2 km. Compared with
the RMS errors in the GPT2w model, those of the HZWD model are decreased by 1.4
mm, 0.9 mm, and 1.2 mm within the three height intervals, corresponding to
improvements of about 6%, 6%, and 32%, respectively. The correction performance
improvement from 5 km to 10 km height is particularly evident. Figure 7a shows the
ECMWF ZWD profile and the ZWD profiles of the three models at 12:00 UTC on 1
January, 2015 at a representative grid point (0 °N, 20 °E). More clearly, Figure 7b shows
the differences between the ZWD profiles of the three models and ECMWF ZWD
profile at different heights. It can be seen that HZWD is the most stable model, showing



the best agreement with the ECMWF ZWD data, which is superior to both the GPT2w,
and the UNB3m, models.

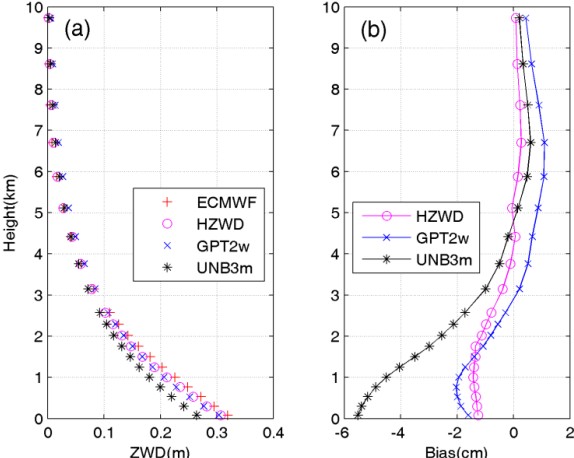


Figure 7 The ZWD profiles of ECMWF and the three models (a) and corresponding biases (b).

The variation of the troposphere has a strong correlation with latitude. To analyse

the correction performances of the three models in different regions around the world,
we calculated the three models' errors in different latitude bands (10 °intervals). Figures
8 and 9 show the correction performances at different latitudes. It can be seen from
Figure 8 that the bias of the UNB3m model is basically positive in the three height
intervals, indicating that its ZWD estimates are relatively large compared to the
ECMWF data. Moreover, the bias in the southern hemisphere is significantly larger than
that in the northern hemisphere, indicating systematic deviations in the southern
hemisphere. Both the GPT2w model and the HZWD model have large biases in the low
latitudes. The biases of the GPT2w model are positive from 2 km to 5 km and 5 km to
10 km height, indicating that the ZWD is overestimated by the GPT2w model with
increasing height. For the HZWD model, the bias in each latitude band is relatively
small with few exceptions, resulting in a global average bias close to zero (see Table 2).

The annual average bias indicates the degree of deviation between the ZWD

estimates of the three models and the reference ECMWF data, while the RMS error
reflects the reliability and stability of the model, *i.e.*, the model precision. It can be seen



from Figure 9 that the precision of HZWD model is significantly better than that of the
UNB3m model across the three height intervals and all latitude bands, which is better
than GPT2w model in general. The precision of the three models declines with
decreasing latitude, because the active change of water vapour in these areas limits the
precision of the model. Corresponding to Figure 8, the errors in UNB3m are asymmetric:
the main reason for this is that the meteorological parameters of UNB3m are
interpolated from the coarse look-up table with a latitude interval of 15 ° and UNB3m
does not consider the longitudinal variations of any meteorological elements. It should
be pointed out that the UNB3m model is based on the simple symmetric assumption of
the northern and southern hemispheres, and its modelling data source only comes from
the atmospheric data collected over North America, which leads to poor precision in
the southern hemisphere, especially in the high latitudes thereof.
Summarising the distributions of bias and RMS error across different latitude
bands, we can see that the HZWD model performs best with the ECMWF data as
reference values. Compared with the models GPT2w and UNB3m, the HZWD model
basically eliminates systematic error in the 5 km to 10 km height interval and the
correction performance is stable at all heights and regions. To investigate the model's
performance over time, the Figure 10 shows the time series of biases for the three
models at 6-hour intervals throughout the year 2015 at grid point (0 °N, 20 °E). We can
see that the HZWD model has the best overall performances within the three height
intervals over the year 2015. We noticed the significantly large biases for all three
models across all three height intervals around the doy 19 and doy 195 of 2015. This
can be attributed to the sharp short-term ZWD variations in the equator area. The short-
term variations are hardly accounted for by all three models which only consider the
seasonal variations of ZWD. Moreover, the GPT2w model has the worst performance
from 5 km to 10 km height, showing significant overestimates of the ZWD. The poor
performance of GPT2w at high heights in the equator area is also identified by Figure
8 and Figure 9.





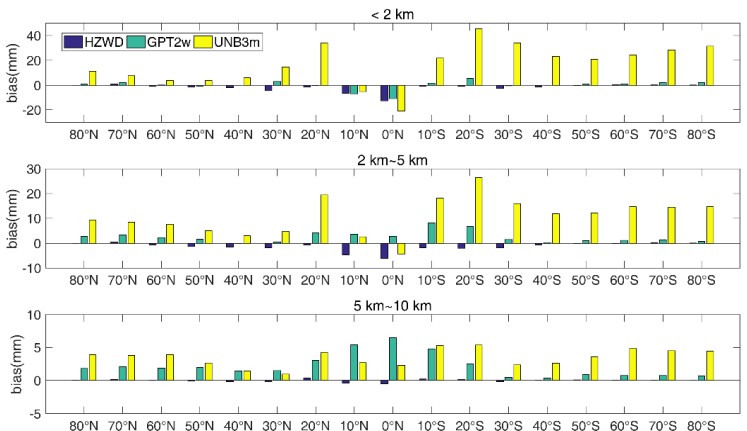


Figure 8 Bias comparisons between the three models in different latitude bands.

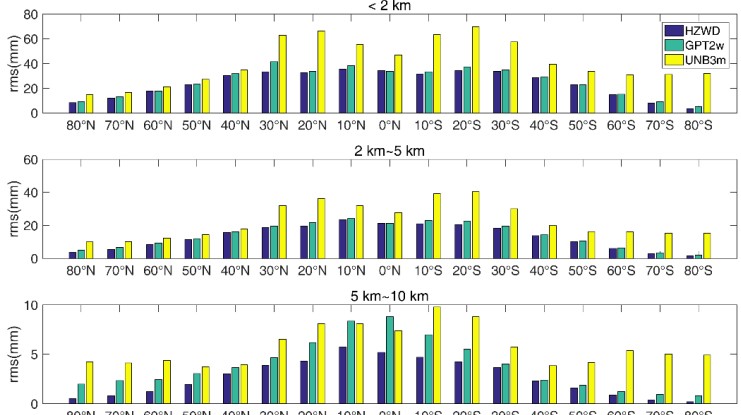


Figure 9 RMS error comparisons between the three models in different latitude bands.





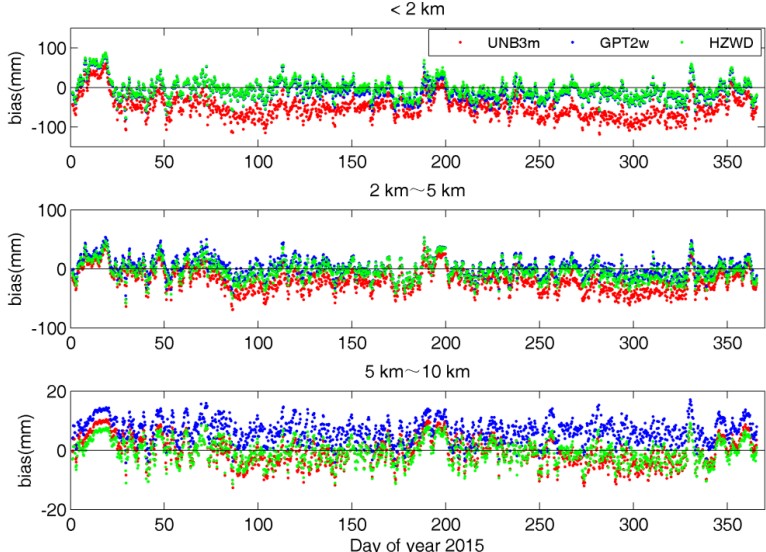

Figure 10 Biases in ZWD estimates of the three models compared to the ECMWF data over the

year 2015 at grid point (0 °N, 20 °E).

## 4.2 Validation with radiosonde data

A radiosonde is used in a sounding technique that regularly releases balloons to collect atmospheric meteorological data at different heights: it can obtain profiles of various meteorological data with high accuracy. At present, the Integrated Global Radiosonde Archive (IGRA) website (ftp://ftp.ncdc.noaa.gov/pub/data/igra/) provides free downloads of global radiosonde data. We used radiosonde data from 318 stations collected in 2015 to test the HZWD model. After data pre-processing, the data with gross errors have been removed and a total of 163,671 radiosonde data epochs remained. With the provided profiles of geopotential height, temperature, and water vapour pressure, the data form of the radiosonde data are very similar to the ECMWF pressure level data, thus the radiosonde ZWDs can be calculated using the same method by equation (2). Before the validation, we conducted an assessment of the uncertainty of ZWD derived from radiosonde data. Rozsa (2014) showed that the uncertainty of ZWD is ±1.5 mm in case of the Vaisala RS-92 radiosondes in Central and Eastern Europe. However, this uncertainty is only valid for the ZWD calculated from the height of





lowest layer and is limited to Europe area. Using the same uncertainties of radiosonde
meteorological data given by the technical specification of the radiosonde (Vaisala 2010)
and the algorithm proposed by Rozsa (2014), we calculated the ZWD uncertainty for
all heights in all radiosonde stations. Figure 11 shows the uncertainty of ZWD with
respect to the height for radiosonde station 01241 located in Orland, Norway
(63.70 °N/9.60 °E/10 m). We can see that the uncertainty of ZWD is less than $\pm 1.5$ mm
near height of 0 m and decrease quickly with increasing height. The global mean
uncertainties of ZWD of all stations in the three height intervals are $\pm 1.3$ mm, $\pm 0.7$
mm and $\pm 0.2$ mm, respectively, indicating the high accuracy of ZWD derived from
radiosonde data.

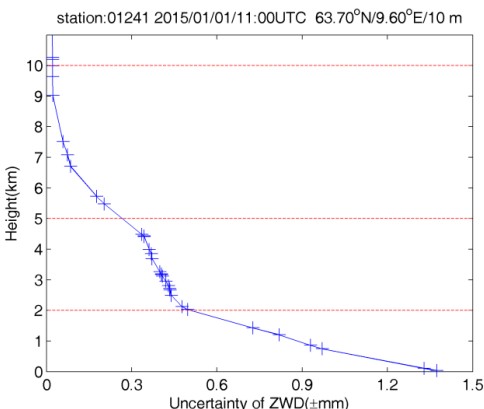


Figure 11 Uncertainty of ZWD with respect to height at station 01241.

Taking the radiosonde ZWDs as reference ZWD values, we validated the ZWDs

from models HZWD, GPT2w and UNB3m. Table 3 shows the statistical results of the
three models. It can be seen from Table 3 that the HZWD model has the best overall
stability of the average bias and RMS error indicating the best precusion, and the
UNB3m model is the worst. Compared with the GPT2w model, the RMS errors in
HZWD in the three height intervals are reduced by 0.6 mm, 0.9 mm, and 1.7 mm, which
equates to precision improvements of 2%, 5%, and 33%, respectively. Taking the
uncertainty of radiosonde ZWD into account, the improvement of HZWD model over
GPT2w model below 2 km seem to be insignificant. However, the validation is based





on the same radiosonde ZWD values and the RMS error of ZWD of HZWD is smaller,
thus we can reasonably expect that the ZWD of HZWD is closer to true ZWD value
than the ZWD of GPT2w in spite of the uncertainty of radiosonde ZWD. It is worth
noting that the bias and RMS error of the HZWD model and the GPT2w model are both
larger than those of the results from ECMWF data in Table 2. The reason is that the
HZWD model and the GPT2w model are based on ECMWF data, thus the test results
with radiosonde data are slightly worse than those using ECMWF data. On the contrary,
the bias of the UNB3m model decreases, and the RMS error between 2 km and 5 km,
and 5 km and 10 km, are less than those in Table 2. It may be due to the fact that most
of the radiosonde stations are in the northern hemisphere, accounting for more than 60%
(192/318) of the total, which has a positive impact on the test results for UNB3m model
based on North American meteorological data.

Figure 12 shows the global distributions of bias for the three models within the

three height intervals, and Figure 13 shows the global distributions of RMS error for
the three models. As can be seen from Figure 12, the three models show a poorer
performance in low-latitude areas than in mid- and high-latitude areas for all height
intervals, similar to the results of in Section 4.1. Within the 5 km to 10 km interval, the
bias of the GPT2w model is large and positive in the equatorial region, indicating that
the ZWD of the GPT2w in this height is significantly overestimated, and the global bias
of the UNB3m model in this height interval is positive, also indicating an overestimate
of the ZWD in the UNB3m model. The bias of the HZWD model does not show obvious
regional differences with respect to height, and the overall distribution of HZWD model
bias has no tendency to either the positive or negative. Figure 13 further illustrates the
precision of the HZWD model. The global RMS error distributions of HZWD model
are similar to that of GPT2w model below 2 km and between 2 km and 5 km, but the
precision of the HZWD model is slightly better. Combining this with the bias
distribution of the GPT2w model in Figure 12, the GPT2w model also has a large RMS
error near the equator in the 5 km to 10 km interval, which shows that the GPT2w model
is unstable at high height in low-latitude areas. The precision of the UNB3m model is
poorer than that of both the HZWD, and GPT2w, models. Below 2 km, the UNB3m



model reaches decimetre-level precision near the equator, and even exceeds 12 cm in
some areas: the distribution of north-south heterogeneity remains obvious.
Table 3 Error statistics for the three models validated by 2015 radiosonde data (unit: mm).

| | < 2 km | | 2 km to 5 km | | 5 km to 10 km | |
|---|---|---|---|---|---|---|
| | bias | RMS | bias | RMS | bias | RMS |
| HZWD | -3.6 | 30.1 | -2.0 | 15.8 | 0.1 | 3.5 |
| GPT2w | -3.2 | 30.7 | 3.5 | 16.7 | 3.3 | 5.2 |
| UNB3m | 5.9 | 46.0 | 6.2 | 23.1 | 2.6 | 5.7 |


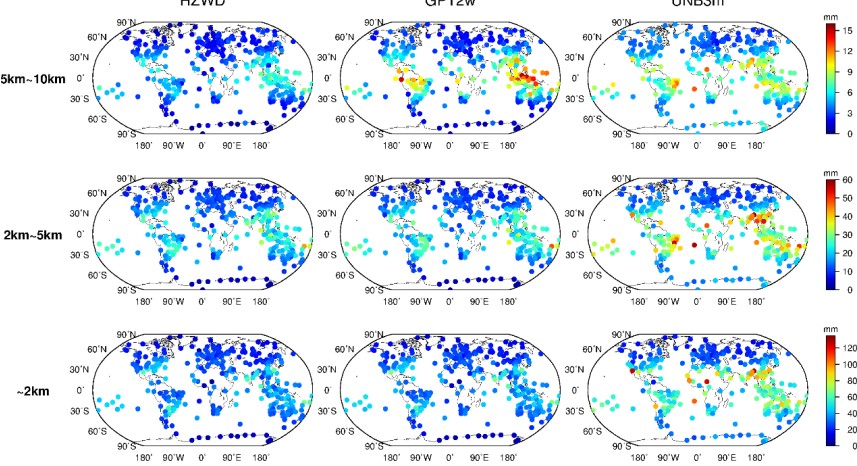

Figure 12 Global distributions of bias for the three models compared to 2015 radiosonde data.





Figure 13 Global distributions of RMS error for the three models compared to 2015 radiosonde

data.

These results validate the spatial stability of the precision of the HZWD model,

furthermore the temporal stability of the model precision is verified next. Figure 14
shows the results of ZWD corrections of the three models for the radiosonde station
01241 for the whole of 2015. It can be seen from Figure 14 that the HZWD model and
the GPT2w model are relatively stable throughout the year, while the correction
performance of the UNB3m model in 2015 is worse than those of the HZWD and
GPT2w models. The probable reason for this is that the UNB3m model only takes into
account the annual variations in the metrological elements with a fixed phase, resulting
in precision instability throughout the year. The improvement performance arising from
use of the HZWD model, compared to that arising from use of the GPT2w model, is
more apparent with increasing height: this shows that modelling ZWD piecewise with
height can effectively approximate the real ZWD profile and improve the precision of
ZWD estimation.

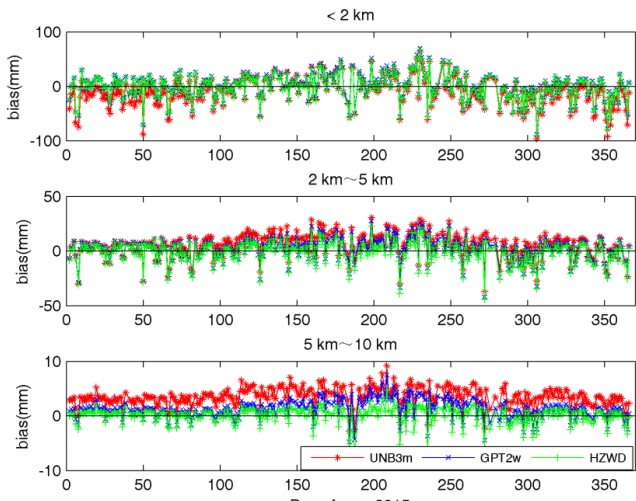


Figure 14 Biases in ZWD estimates of the three models for radiosonde station 01241 over the

427                                               year 2015.




## 5 Conclusions

The complexity of spatio-temporal variations makes the modelling of tropospheric ZWD difficult. In this paper, the characteristics of vertical variation of wet delay are analysed. The troposphere is divided into three height intervals: below 2 km, 2 km to 5 km, and 5 km to 10 km according to different trends (10 km is assumed to represent the empirical tropopause). A quadratic polynomial and two exponential functions are used to describe the variation of wet delay within each of the three intervals. Based on the monthly mean data of ECMWF ZWD from 2001 to 2010, a global ZWD model with spatial resolution of $5\,° \times 5\,°$ was established with height fitting followed by periodic fitting. Using the ECMWF ZWD data for 2015, the annual average RMS errors in the HZWD model are 23.8 mm, 13.1 mm, and 2.6 mm in the below 2 km, 2 km to 5 km, and 5 km to 10 km height intervals, respectively, which is far superior to the performance of the UNB3m model. Compared to the currently most accurate wet delay empirical model (the GPT2w model), the precisions within the three height intervals improved by 6%, 6%, and 32%, respectively. The testing results of radiosonde data from 318 stations in 2015 show that the annual average RMS errors of the HZWD model are 30.1 mm, 15.8 mm, and 3.5 mm, which are 2%, 5%, and 33% better than those of the GPT2w model, respectively. Compared with the GPT2w, and UNB3m, models, the HZWD model offers the highest spatio-temporal stability.

The HZWD model offers good precision stability in the vertical direction and can meet the requirements of ZWD correction at different heights within the troposphere; however, it can be seen that neither the HZWD, nor the GPT2w, models, *i.e*., those non-meteorological parameter-based models, performed well in the lower region of the troposphere. In addition, compared with the GPT2w model, HZWD model is a closed model with a limitation to facilitate on-site meteorological observations. Further research is required to assess the variation in and factors influencing of the wet delay and explore the possibility of incorporation of on-site meteorological data.

*Acknowledgements:* The authors would like to thank the ECMWF and IGRA for





providing relevant data. This research was supported by the National Key Research and
Development Program of China (2016YFB0501803) and the National Natural Science
Foundation of China (41574028) and Key Laboratory of Geospace Environment and
Geodesy, Ministry of Education, Wuhan University (16-02-03).

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
