# Peer review of "An empirical zenith wet delay correction model using"

_Annales Geophysicae, 2018_

## Referee Comment (RC1)

Review of:

YiBin Yao and YuFeng Hu, An empirical zenith wet delay correction model using piecewise height functions

The manuscript presents global model of the tropospheric delay part that is caused by wet component (water vapour). It combines the idea of global zenith wet delay model that requires non meteorological parameters with the focus on the behavior of the delay in height. The base data of temperature, geopotential height and specific humidity (source: ERA-Interim) for 37 pressure levels (monthly mean for years 2001-2010) are used to obtain a time series of Zenith Wet Delay that are further processed. The proposed model concerns 3 layers each of which is treated with separate global grid of ZWD for bottom of the layer and separate set of coefficients for change of ZWD in height. For each layer the proposed model assumes separate functions for change of ZWD in height (quadratic polynomial for layer <2km and exponential for both 2 to 5 km and 5 to 10 km), resulting in a set of 27 parameters for each grid point. Next the model is validated against ZWD profiles obtained from ERA-Interim data for year 2015 (6 hours resolution) and radiosonde data (318 stations, year 2015) and compared to GPT2w and UNB3m empiric models.

The concept is interesting and pushes the tropospheric modeling for space geodetic techniques a bit further. The final improvement reaches about 1-1.5 mm when comparing to other empiric models. The manuscript is well organized. The analysis vertical variations of ZWD and creation of the model takes about 35% of the contents while detailed validation and comparison with GPT2w and UNB3m of the model takes about 40%. However, the context and motivation could be improved. There are also some inconsistencies across the contents. Below are the remarks that could improve the clearness of the contents and substantiality of the conclusions.

General remarks:

1. In reviewers opinion, the reader does not get clear insight into current state of tropospheric modeling after reading the introduction. In addition, the context could contain the view of the empiric models and their quality in broader scope of model types that are commonly used in space geodetic techniques. The title suggests that the scope concerns empiric modeling, however, It could be clearly stated in the text (abstract or introduction).
2. The contents would benefit of clear statement, analysis, whether using meteorological parameters is not as efficient as using set of ZWD and coefficients for height correction.
3. The manuscript should contain the discussion on the data size that the model consist of with comparison to other models. (P11.220)
4. The ZWD as well as its typical modeling error decreases with height reaching small values for top layers. In addition, The variation there is small, which was also shown in the contents (truncating of semi-annual terms for middle and top layers, page 10). The contents would benefit of some reasoning if the improvement is significant facing the above against application in space geodetic techniques that are mentioned in the contents as possible applications (wide area augmentation systems, real-time aircraft navigation and positioning).

Detailed remarks:

P2.56 Citation needed here

P3.61 The reader gets confused if using meteo data or ZWD directly influences the precision of the resulting model.

P3.73-75 Does the quality of the models mentioned here brings any positioning deterioration when comparing to quality of the proposed model?

P4.90-93 Is the model proposed in the manuscript in fact comprises of the profiles of coefficients and ZWD. Is it more difficult to attach the profiles of meteo parameters and calculate the ZWD in the model procedure? The reasoning here is not convincing.

P5.125-127 and P6.152 If the conclusion is that the change is linear, why the model uses quadratic function below 2km?

P9.187 Figure 5 could be more informative if fitting curves with annual term were added in context of p10.198-203

P11.220 The statement here should be extended with comparison to other empiric models

Technical remarks:

P4. 107 It is not necessary to repeat the description of the symbols that are already described at the same page

P18.367 precusion -> precision

P14.296-298 The sentence here is not necessary as bias and RMS are already described at  P12.237-238.

--

10.07.2018, Jakub Z Kalita

---

## Referee Comment (RC2)

Review of the paper

MS No.: angeo-2018-49

Title: An empirical zenith wet delay correction model using piecewise height functions

**General comment**

The paper presents a climatology of the ZWD field by modelling the spatial, temporal and vertical variations of this field. With respect to state-of the art models such as GPT2w, the novelty of the paper is the modelling of the vertical variation of the ZWD by dividing the troposphere into three different regions (0-2 km; 2-5km and 5-10km) and adjusting different functions for each region (quadratic polynomials in the first case and exponentials in the last two cases).

The topic is of interest as the wet path delay is a major error source in various remote sensing techniques such as satellite altimetry, as well as in satellite positioning and navigation.

The paper is scientifically sound and well structured. Although the paper is generally well written, some parts, particularly those concerning the discussion of the results, need some language improvement and polishing.

The less strong points of the paper are the need for some more details and clarification in some parts and the fact that the proposed model is only a marginal improvement with respect to GPT2w and that, considering the large variability of the ZWD field, improvements of millimetre level are insignificant. Nevertheless, the proposed methodology is overall correct and interesting.

Below I point out some suggestions for improvement.

**Detailed comments**

General comment to all figure captions: add to captions the necessary information to understand what each figure represents, e.g. the location of the represented point, etc. Each figure should be self-explanatory.

The plots of ZWD against height are more intuitive if ZWD is given in x-axis and height in the y-axis. Although not mandatory, please consider changing the plots accordingly.

**1. Introduction**

The authors should distinguish between the various types of ZWD models, namely:

(1) Models that make use of observations (e.g. radiosonde profiles) or 3D fields from atmospheric models (e.g. ECMWF) such as the Davis et al, 1995, Eq. (1). These provide the best accuracy as they model the spatial, temporal and vertical variations of the field.

(2) Models that make use of single layer parameters such as total column water vapour (TCWV) and temperature, such as those by Bevis et al. 1992, 1994. See for example, Fernandes et al., 2013 for a reference to the use of this model and that of Stum et al., 2011 which only uses TCWV. These provide similar results to those in (1) but only at the level of the model orography to which the meteorological parameters refer to. Note that TCWV includes the modelling of the amount of water vapour along the vertical profile up to the reference orography, so the result is not very different from case (1). As this orography may depart significantly from the actual surface and the vertical variation of the ZWD is not well known, at a different surface elevation they possess errors associated with the uncertainty in the modelling of the ZWD height variation (see for example Fernandes et al, 2014, Vieira et al,. 2017).

(3) Models such as Saastamoinen or Hopfield, that make use only of surface observations, these lack information about the vertical distribution of the water vapour, so they are considerably less accurate than (1) and (2)

(4) Climatologist which, unlike the other mentioned approaches, do not use observations nor meteorological parameters from atmospheric models; they attempt to model the space-time and sometimes also the vertical variation of the ZWD field, such as the GPT2w type of models.

Clearly, the paper proposes a model that belongs to the last type. The paper would benefit from a clarification of this nature, putting this study into the right context. Moreover, the accuracy of each of the two climatology models used in the paper for comparison with the new model (GPT2w, and UNB3m)) should be clearly indicated as quoted by the respective authors (e.g. 3.6 cm for GPT2w)

The paper should include a brief description of the space-time variability of the ZWD field as well as of its vertical variation, as published in the literature, including proper references. The vertical variation is mentioned in section 2, but the space-time variations are missing. About the vertical variation, the work by Kouba, 2008 is worth mentioning.

The use of the words "obvious" and "obviously" is not recommend in scientific writing. I suggest to replace them throughout the text by "clear" or similar synonyms.

**2. Vertical variations of ZWD**

In Eq. (1) the authors quote the values for the refractivity constants as given by Bevis et al, 1992. However, Bevis et al, 1994 revisited these constants. I recommend the addition of this reference (see list of some suggested references at the end of this review) and of the updated values for the constants ($k_3$=3.739 x$10^5$ K$^2$mbar$^{-1}$; $k_2'$ =22.1 Kmbar$^{-1}$).

Authors should justify the use of "pressure levels" instead of "model levels" as, according to ECMWF, for wet path delay computations the second should be adopted as they lead to better accuracy.

Although ECMWF allows the extraction of ERA Interim with various spatial resolutions, the actual spatial resolution of the model is 0.75°× 0.75° (regular grid) or about 80 km (Gaussian grid), Dee et al., 2011. Please rephrase the sentence describing the model accordingly.

Equation (2): a reference should be provided for the first line of Eq. (2).

Caption of Figure 1 – Please add time and place to which profiles correspond.

Caption of Figure 2 – Please explain how these figures were made. Points represent mean values of ZWD gradients for which period and location?

The sentence "*the ZWD gradients at high latitudes are much larger and water vapour is more variable than at low latitudes, resulting from the fact that the water vapour at high latitudes are more variable*" is not true and needs rephrasing. Please see for example figures 1 and 2 in Fernandes et al, 2013 for the mean and standard deviation of the ZWD. You can see that the

ZWD variability is largest in the tropics and minimum at high latitudes. In the paper you probably refer to the variation of ZWD with height. Even in this case the sentence is also not true as the scales in the various plots in figures 3 and 4 are different. Please use the same scale in all figures, so that they can be compared. Some additional background information about the ZWD variability (in space, time and vertically) by citing references such as the one mentioned above should be given at the introduction.

In "and semi-annual, cycles" please remove the ",".

Table1 – I don't understand what is represented in the first line of the table. According to the text, the piecewise height functions statistics should be the same to those for the quadratic functions for the layer 0-2km and to the exponential functions in the other two cases. Please explain.

**3 The HZWD model**

Figure 6 – for comparison of the various plots the sane colour scale should be used. In this way the decrease of ZWD with height would be clearer.

"For each grid point, there are 27 parameters" – please indicate explicitly the 27 parameters, e.g. 5 for $Z_1$, etc.

Please avoid using the same symbol for different variables in different equations, e.g. $\beta$.

**4 Validation and analysis of the HZWD model**

In my view, the sub-section on the validation using ECMWF ERA Interim data for the year 2015 does not add any information to the validation performed with radiosondes. Indeed, the comparison with ECMWF data from a different period is not a true validation. Since the HZWD model only uses periodic functions (annual and semi-annual), examining the differences between model predictions and ECMWF-derived ZWD for a year inside or outside the 10-year period used in model fitting should give very similar results. This would not be the case if e.g. the HZWD model included inter-annual signals. Please try to explain the utility of including this analysis or consider removing it from the paper.

Please define RMS.

In the sentence "In equation (5) and (6), $ZWD_i^M$ is the value estimated by the model and $ZWD_i^0$ is the reference value.", clarify the sentence by writing "In equation (5) and (6), $ZWD_i^M$ is the value estimated by the HZWD model developed in this study and $ZWD_i^0$ is the reference value."

In Eq. (7) please specify full expressions and how each term is computed: $T_m$, $g_m$, etc.

In Eq. (8) please define all terms and describe how they are computed.

The equivalent to Figure 10 but for the RMS instead of the bias should be added. This would give a better indicator of each model accuracy. The same should be done for the comparison with radiosondes (Figure 14)

Replace "precusion" by "precision"

The sentences "Taking the uncertainty of radiosonde ZWD into account, the improvement of HZWD model over GPT2w model below 2 km seem to be insignificant. However, the validation is based on the same radiosonde ZWD values and the RMS error of ZWD of HZWD is smaller, thus we can reasonably expect that the ZWD of HZWD is closer to true ZWD value than the ZWD of GPT2w in spite of the uncertainty of radiosonde ZWD." are confusing and repetitive. Please rephrase the second sentence and try to split into shorter sentences.

**5. Conclusions**

Mentioning the percentage of improvement for the various layers may be misleading as e.g. 33% of the signal for the top layer is only 2-3 mm, which is insignificant. Authors should put their results into perspective of the magnitude of the ZWD field in each layer.

The last paragraph of the paper is an example of the need for language improvement:

"The HZWD model offers good precision stability in the vertical direction and can meet the requirements of ZWD correction at different heights within the troposphere; however, it can be seen that neither the HZWD, nor the GPT2w, models, i.e., those non- meteorological parameter-based models, performed well in the lower region of the troposphere. In addition, compared with the GPT2w model, HZWD model is a closed model with a limitation to facilitate on-site meteorological observations. Further research is required to assess the variation in and factors influencing of the wet delay and explore the possibility of incorporation of on-site meteorological data."

Suggestions:

Remove "," in "GPT2w, models

Replace "lower" by "lowest"

Replace "compared with the GPT2w model, HZWD model is a closed model with" by "compared with GPT2w, HZWD is a closed model with", this way avoiding the repetition of the word "model".

Remove "and" in "in and factors"

List of suggested references:

Bevis, M.; Businger, S.; Chiswell, S.; Herring, T. A.; Anthes, R. A.; Rocken, C.; Ware, R. H. (1994). GPS meteorology—Mapping zenith wet delays onto precipitable water. J. Appl. Meteorol., 33, 379–386.

Dee, D. P., Uppala, S. M., Simmons, A. J., Berrisford, P., Poli, P., Kobayashi, S., Andrae, U., Balmaseda, M. A., Balsamo, G., Bauer, P., et al. (2011). The ERA-Interim reanalysis: Configuration and performance of the data assimilation system. Q. J. R. Meteorol. Soc., 137, 553–597.

Fernandes, M. J., Nunes, A. N., & Lázaro, C. (2013). Analysis and Inter-Calibration of Wet Path Delay Datasets to Compute the Wet Tropospheric Correction for CryoSat-2 over Ocean. Remote Sensing, 5(10), 4977-5005. doi:10.3390/rs5104977

Fernandes, M. J., Lázaro, C., Nunes, A. N., & Scharroo, R. (2014). Atmospheric Corrections for Altimetry Studies over Inland Water. Remote Sensing, 6(6), 4952-4997. doi:10.3390/rs6064952

Kouba, J. (2008). Implementation and testing of the gridded Vienna Mapping Function 1 (VMF1). J. Geodesy, 82, 193–205.

Stum, J.; Sicard, P.; Carrere, L.; Lambin, J. (2011). Using objective analysis of scanning radiometer measurements to compute the water vapor path delay for altimetry. IEEE Trans. Geosci. Remote Sens., 49, 3211–3224.

Vieira, T., Fernandes, M. J., Lázaro, C. (2017). Analysis and retrieval of tropospheric corrections for CryoSat-2 over inland waters. Advances in Space Research. doi:10.1016/j.asr.2017.09.002

2 August 2018

Joana Fernandes

---

## Author Comment (AC3) · 30 Aug 2018

The comment was uploaded in the form of a supplement:
https://www.ann-geophys-discuss.net/angeo-2018-49/angeo-2018-49-AC3-supplement.pdf

---

## Author Response (AR1)

**Responses to Referees' and Editor's Comments (angeo-2018-49)**

We thank both the referees and editor for their insightful comments and constructive suggestions. We have addressed all their comments in the revised manuscript. Below are our responses to the critical comments (*Italics*). The page and line numbers in our responses refer to those in the revised manuscript.

**Responses to Referee #1:**

1. In reviewers opinion, the reader does not get clear insight into current state of tropospheric modeling after reading the introduction. In addition, the context could contain the view of the empiric models and their quality in broader scope of model types that are commonly used in space geodetic techniques. The title suggests that the scope concerns empiric modeling, however, It could be clearly stated in the text (abstract or introduction).

**Authors:** To make the introduction more clearly, we rewrote this part to clarify the tropospheric models into 4 types: 1. Ray tracing 2. Single layer parameterization 3. Surface observation parameterization 4. Empirical and climatological model. The proposed model HZWD belongs to the fourth type. In addition, we added the description of the qualities of the empirical models. (Page 2, Line 48 to Page 3, Line 77)

**2. The contents would benefit of clear statement, analysis, whether using meteorological parameters is not as efficient as using set of ZWD and coefficients for height correction.**

**Authors:** GPT2w model is the representative empirical model using meteorological parameters. In this paper, we have compared the proposed HZWD model with GPT2w model. We added the comparison about data size of the model parameters for these two models. (Page 13, Lines 251-255) Combining with the precision validations, the HZWD model using set of ZWD and coefficients is more efficient than the GPT2w model using meteorological parameters. (Page 25, Lines 504-506)

**3. The manuscript should contain the discussion on the data size that the model consist of with comparison to other models. (P11.220)**

**Authors:** We added the discussion about data size of the model parameters. (Page 13, Lines 251-255)

4. The ZWD as well as its typical modeling error decreases with height reaching small values for top layers. In addition, The variation there is small, which was also shown in the contents (truncating of semi-annual terms for middle and top layers, page 10). The contents would benefit of some reasoning if the improvement is significant facing the above against application in space geodetic techniques that are mentioned in the contents as possible applications (wide area augmentation systems, real-time aircraft navigation and positioning).

**Authors:** In the application of space geodetic techniques (wide area augmentation systems, realtime aircraft navigation and positioning), the zenith delay errors directly relate to the errors in the height estimates in the positioning (B öhm and Schuh, 2013). For instance, above height of 5 km, the improvement of HZWD model over GPT2W model is about 2 mm, which will result in about 4 mm improvement of height estimates in the above applications. This magnitude of the improvement is quite significant for positioning. We added the corresponding discussions in the paper. (Page 15, Lines 319-321; Page 20, Line 419 to Page 21, Line 421)

**P2.56 Citation needed here**

Authors: We added a reference to Leandro (2006). (Page 3, Line 68)

**P3.61 The reader gets confused if using meteo data or ZWD directly influences the precision of the resulting model.**

**Authors:** The TropGrid model expresses ZWD as a function of water vapor pressure and weighted mean temperature, which involves two variable quantities with individual uncertainties. In TropGrid2, ZWD is directly modeled as an empirical exponential expression with only one parameter, which reduce the uncertainties. To avoid the confusion, we rewrote this sentence as "The improved TropGrid2 model (Sch üler, 2014) enhances the efficiency of ZWD calculation by directly modelling ZWD with the exponential function". (Page 3, Lines 70-72)

**P3.73-75* Does the quality of the models mentioned here brings any positioning deterioration when comparing to quality of the proposed model?**

Authors: According to the B öhm and Schuh (2013), the zenith delay error will result in two times errors in the station height estimates in positioning (Newly added sentence: Page 3, Line 87 to Page 4, Line 88). The commonly used empirical models such as UNB3m and GPT2w has a limitation in ZWD estimation at high heights due to the simple approximations of ZWD vertical variation. The HZWD model has the best accuracy of ZWD estimation especially at high heights, which will improve the positioning precision.

**P4.90-93 Is the model proposed in the manuscript in fact comprises of the profiles of coefficients and ZWD. Is it more difficult to attach the profiles of meteo parameters and calculate the ZWD in the model procedure? The reasoning here is not convincing.**

**Authors:** HZWD is an empirical climatological model using piecewise functions to directly describe the ZWD vertical variations in different height intervals. We rewrote the sentences as "With the profiles of water vapour pressure and temperature, one can obtain the accurate ZWD by ray tracing method. However, in practical applications (such as aircraft navigation and positioning and wide area augmentation), we usually uses empirical models for ZWD corrections due to the unavailability of meteorological data profiles. Therefore, it is necessary to develop an empirical ZWD model with high precision". (Page 4, Lines 104-109)

**P5.125-127 and P6.152 If the conclusion is that the change is linear, why the model uses quadratic function below 2km?**

Authors: Figure 2a shows the changes of **ZWD vertical gradients** with respect to the height and Figure 2b gives the linear fitting of ZWD gradients below 2 km. The linear ZWD vertical gradient is the derivative of ZWD along vertical direction (i.e.,  $\frac{\partial ZWD}{\partial h}$ ), thus the ZWD could be characterized using quadratic function. We added a description about the ZWD vertical gradients. (Page 5, Lines 137-139)

*P9.187 Figure 5 could be more informative if fitting curves with annual term were added in context of p10.198-203*

Authors: The fitting curves of the figure 5 are (w= $2\pi/365.25$ , x=doy): z1=0.2911+0.0237\*cos(w\*x)+0.0312\*sin (w\*x)- 0.0006\*cos(2w\*x)-0.0227\*sin (2w\*x) z2=0.1215+0.0118\*cos(w\*x)+0.0203\*sin (w\*x)+ 0.0004\*cos(2w\*x)-0.0146\*sin (2w\*x) z3=0.0255+0.0031\*cos(w\*x)+0.0070\*sin (w\*x)- 0.0019\*cos(2w\*x)-0.0044\*sin (2 w\*x) We added the interpretation in the context as you suggested as "The fittings show that the annual means, and annual, and semi-annual amplitudes of z1, z2 and z3 are distinct. For instance, the cycle fitting results at a grid (0 °N, 0 °E) (Figure 5) indicate that the temporal parameters (i.e., A0, A1, B1, A2, and B2) of z1 are 0.2911 m, 0.0237 m, 0.0312 m, -0.0006 m, and -0.0227 m, respectively; the temporal parameters of z2 are 0.1215 m, 0.0118 m, 0.0203 m, 0.0004 m, and -0.0146 m, respectively; the temporal parameters of z3 are 0.0255 m, 0.00031 m, 0.0070 m, -0.0019 m, and -0.0044 m, respectively." (Page 11, Lines 217-223)

*P11.220 The statement here should be extended with comparison to other empiric models* **Authors:** See previous response. We added the comparison. (Page 13, Lines 251-255)

P4. 107 It is not necessary to repeat the description of the symbols that are already described at the same page

Authors: We deleted the repeatable description of the symbols *e* and *T*.

*P18.367 precusion -> precision* **Authors:** Done. (Page 20, Line 416)

P14.296-298 The sentence here is not necessary as bias and RMS are already described at P12.237-238.

Authors: We deleted this sentence from the paper. (Page 17, Line 346)

**Responses to Referee #2:**

General comment to all figure captions: add to captions the necessary information to understand what each figure represents, e.g. the location of the represented point, etc. Each figure should be self-explanatory.

**Authors:** We rewrote all figures captions to make them more self-explanatory as you suggested (Figures 1-14 in the paper).

The plots of ZWD against height are more intuitive if ZWD is given in x-axis and height in the yaxis. Although not mandatory, please consider changing the plots accordingly.

**Authors:** As you suggested, we switched the x-axis and y-axis of some plots (Page 6, Figures 1-2; Page 8, Figures 3-4) to make them more intuitive.

The authors should distinguish between the various types of ZWD models, namely:

(1) Models that make use of observations (e.g. radiosonde profiles) or 3D fields from atmospheric models (e.g. ECMWF) such as the Davis et al, 1995, Eq. (1). These provide the best accuracy as they model the spatial, temporal and vertical variations of the field.

(2) Models that make use of single layer parameters such as total column water vapour (TCWV) and temperature, such as those by Bevis et al. 1992, 1994. See for example, Fernandes et al., 2013 for a reference to the use of this model and that of Stum et al., 2011 which only uses TCWV. These provide similar results to those in (1) but only at the level of the model orography to which the meteorological parameters refer to. Note that TCWV includes the modelling of the amount of water vapour along the vertical profile up to the reference orography, so the result is not very different from case (1). As this orography may depart significantly from the actual surface and the vertical variation of the ZWD is not well known, at a different surface elevation they possess errors associated with the uncertainty in the modelling of the ZWD height variation (see for example Fernandes et al, 2014, Vieira et al., 2017).

(3) Models such as Saastamoinen or Hopfield, that make use only of surface observations, these lack information about the vertical distribution of the water vapour, so they are considerably less accurate than (1) and (2)

(4) Climatologist which, unlike the other mentioned approaches, do not use observations nor meteorological parameters from atmospheric models; they attempt to model the space-time and sometimes also the vertical variation of the ZWD field, such as the GPT2w type of models.

Clearly, the paper proposes a model that belongs to the last type. The paper would benefit from a clarification of this nature, putting this study into the right context. Moreover, the accuracy of each of the two climatology models used in the paper for comparison with the new model (GPT2w, and UNB3m)) should be clearly indicated as quoted by the respective authors (e.g. 3.6 cm for GPT2w) **Authors:** We accepted your suggestions and rewrote the introduction to clarify the four types of ZWD models and clearly indicated the accuracies of the GPT2w and UNB3m models (Page 2, Line 48 to Page 3, Line 77).

The paper should include a brief description of the space-time variability of the ZWD field as well as of its vertical variation, as published in the literature, including proper references. The vertical variation is mentioned in section 2, but the space-time variations are missing. About the vertical variation, the work by Kouba, 2008 is worth mentioning.

**Authors:** We added a description of the space-time variability of the ZWD as "Its spatial distribution is characterized with a near-zonal dependency, with values varying from about 2 cm at high latitudes to about 35 cm near the equator (Fernandes et al., 2013). The temporal variation pattern of ZWD is mainly characterized by the seasonal variability including annual and semi-annual components (Jin et al., 2007; Nilsson et al., 2008). The high viabilities in ZWD make itself the main factor influencing tropospheric delay correction."(Page 2, Lines 41-47). About vertical variation of ZWD, we added a sentence as "Kouba (2008) proposed an empirical exponential model to account for the height dependency of ZWD, but it only be applicable within the height below 1000 m."(Page 3, Lines 80-82).

The use of the words "obvious" and "obviously" is not recommend in scientific writing. I suggest to replace them throughout the text by "clear" or similar synonyms.

**Authors:** We replaced the words by other synonyms (e.g., clear, evident) throughout the paper as you suggested.

In Eq. (1) the authors quote the values for the refractivity constants as given by Bevis et al, 1992. However, Bevis et al, 1994 revisited these constants. I recommend the addition of this reference (see list of some suggested references at the end of this review) and of the updated values for the constants  $(k_3=3.739 \times 105 \text{ K}^2\text{mbar}^{-1}; k_2 = 22.1 \text{ Kmbar}^{-1}).$

**Authors:** We revised the values for the constants and corresponding reference as you suggested. (Page 4, Lines 101-102)

Authors should justify the use of "pressure levels" instead of "model levels" as, according to ECMWF, for wet path delay computations the second should be adopted as they lead to better accuracy.

**Authors:** The figures below show the water vapour pressure and ZWD derived from pressure levels and model levels data at the representative grid point (0 °N, 0 °E) at 12:00 UTC on 1 January, 2005. From the figures we can see that the model levels-derived water vapour pressure data and ZWD data are denser than that derived from pressure levels above height of 20 km, and the water vapour pressure and ZWD are quite close to zero. **About ZWD comparison, the RMS between these two data is about 0.11 mm**. Considering such small difference, there is almost no difference in ZWD modelling with pressure levels or model levels data in our study, especially the modelling focuses on the height interval below 10 km.

Although ECMWF allows the extraction of ERA Interim with various spatial resolutions, the actual spatial resolution of the model is  $0.75 \approx 0.75 \circ$  (regular grid) or about 80 km (Gaussian grid), Dee et al., 2011. Please rephrase the sentence describing the model accordingly.

**Authors:** We rephrased the sentence as "ERA-Interim can provide data at 0:00, 6:00, 12:00, and 18:00 UTC daily with a spatial resolution of not more than  $0.75 \circ \times 0.75 \circ$  and 37 pressure levels (Dee et al., 2011)." (Page 4, Line 115 to Page 5, Line 117)

*Equation (2): a reference should be provided for the first line of Eq. (2).* **Authors:** We added a reference to B öhm and Schuh (2013). (Page 5, Line 121)

*Caption of Figure 1 – Please add time and place to which profiles correspond.* **Authors:** We added time and place as you suggested. (Page 6, Lines 147-148)

Caption of Figure 2 – Please explain how these figures were made. Points represent mean values of ZWD gradients for which period and location?

**Authors:** We rewrote the caption as "ZWD vertical gradients profile (a) and linear fit with height below 2 km (b) at a grid point (0 °N, 0 °E) at 12:00 UTC on 1 January, 2010." (Page 6, Lines 150-151)

For better understanding, we also added a sentence about ZWD vertical gradients in the context as "Further, the derivative of the ZWD with respect to height (i.e., ZWD vertical gradient) is analyzed to better understand the characteristic of the ZWD vertical distributions." (Page 5, Lines 137-139)

The sentence "the ZWD gradients at high latitudes are much larger and water vapour is more variable than at low latitudes, resulting from the fact that the water vapour at high latitudes are more variable" is not true and needs rephrasing. Please see for example figures 1 and 2 in Fernandes et al, 2013 for the mean and standard deviation of the ZWD. You can see that the ZWD variability is largest in the tropics and minimum at high latitudes. In the paper you probably refer to the variation of ZWD with height. Even in this case the sentence is also not true as the scales in the various plots in figures 3 and 4 are different. Please use the same scale in all figures, so that they can be compared. Some additional background information about the ZWD variability (in space, time and vertically) by citing references such as the one mentioned above should be given at the introduction.

**Authors:** It's a mistake. We corrected the interpretation as "It is worth noting that the ZWD gradients at low latitudes are much larger and water vapour is more variable than at high latitudes, resulting from the fact that the water vapour at low latitudes are more variable." (Page 6, Lines 155-157; Page 8, Figures 3-4). Figure 3 and Figure 3 were replotted with same scale. (Page 8, Figures 3-4). The background information about the ZWD variability were added. (see previous response)

*In "and semi-annual, cycles" please remove the ",".* **Authors:** Done. (Page 11, Line 225)

Table 1 - I don't understand what is represented in the first line of the table. According to the text, the piecewise height functions statistics should be the same to those for the quadratic functions for

**the layer 0-2km and to the exponential functions in the other two cases. Please explain.**

**Authors:** For other scholars such as Song et al. (2011), the single function is used to fit the ZWD variations respect to height **throughout the whole troposphere.** While in our study, we used three functions to fit the ZWD **in three different height intervals**. In the second row of Table 1, it gives the fitting performances of piecewise height functions in three height intervals. To be consistent with the common fitting methods such as Song et al. (2011), single quadratic polynomial is used to fit the ZWD throughout the whole height interval below 10 km. Therefore, the third row of Table 1 shows **the part RMSs of single quadratic polynomial fitting in three height intervals**, which is different from the results of piecewise height functions. For clarity, we rewrote the interpretation as "Table 1 summarises the global fitting statistics of different fit functions, demonstrating the superiority of piecewise height functions to the single polynomial function and single exponential function used for the whole troposphere." (Page 7, Lines 174-176)

Figure 6 – for comparison of the various plots the sane colour scale should be used. In this way the decrease of ZWD with height would be clearer. Authors: Done. (Page 12, Figure 6)

*"For each grid point, there are 27 parameters" – please indicate explicitly the 27 parameters, e.g. 5 for Z1, etc.*

Authors: We rewrote the sentence as "Each grid point contains 7 primary coefficients: z1, a1, a2, z2,

 $\beta_2$ , z3, and  $\beta_3$ . Among these coefficients, z1, z2, z3, a1, and a2 are further expressed by the equation

(4) with 5 temporal parameters, respectively. Therefore, there are 27 parameters for each grid point and total 68094 parameters for the HZWD model". (Page 12, Line 249 to Page 13, Line 252)

Please avoid using the same symbol for different variables in different equations, e.g.  $\beta$ . Authors: We replaced the symbol  $\beta$  with  $\gamma$  in equations (7) (Page 14). The other symbols have the same meanings.

In my view, the sub-section on the validation using ECMWF ERA Interim data for the year 2015 does not add any information to the validation performed with radiosondes. Indeed, the comparison with ECMWF data from a different period is not a true validation. Since the HZWD model only uses periodic functions (annual and semi-annual), examining the differences between model predictions and ECMWF-derived ZWD for a year inside or outside the 10-year period used in model fitting should give very similar results. This would not be the case if e.g. the HZWD model included inter-annual signals. Please try to explain the utility of including this analysis or consider removing it from the paper.

Authors: The HZWD model is established using ERA-Interim monthly mean pressure levels data from 2001 to 2010. In validation, the pressure levels data with time resolution of 6 h over 2015 are used. These two data are different in period and temporal resolution, so we can use the more accurate ECMWF data (with the full temporal resolution of 6 hours) in a different year (2015) to validate the empirical seasonal models. We have emphasized it in the paper as "The establishment of the HZWD model is based on the monthly mean profiles of ERA-Interim pressure levels data from 2001 to 2010, while we used the ERA-Interim pressure levels data with the full time resolution

of 6 hours in 2015 for the model validation". (Page 15, Lines 301-304). To make it more clearly, we added a new sentence as "This is to validate the model performance on the daily scale." (Page 15, Line 307)

**Please define RMS.**

Authors: We added the definition as "root mean square (RMS) error". (Page 13, Line 274)

In the sentence "In equation (5) and (6),  $ZWD_i^M$  is the value estimated by the model and  $ZWD_i^0$  is the reference value.", clarify the sentence by writing "In equation (5) and (6),  $ZWD_i^M$  is the value estimated by the HZWD model developed in this study and  $ZWD_i^0$  is the reference value." **Authors:** Done. (Page 14, Lines 278-279)

In Eq. (7) please specify full expressions and how each term is computed: Tm, gm, etc. Authors: We specified all terms (Page 14, Lines 284-288) and gave the expressions to calculate  $T_m$  and  $g_m$ . (Page 14, Eq. (8) and Eq (9))

**In Eq. (8) please define all terms and describe how they are computed.**

Authors: Since the terms in the equation have been defined in the aforementioned equations in the paper, we choose not to repeat the definitions. As you suggested, we added the descriptions about how to compute the  $T_m$  and  $g_m$  in the equation. (Page 14, Line 298 to Page 15, Line 299)

The equivalent to Figure 10 but for the RMS instead of the bias should be added. This would give a better indicator of each model accuracy. The same should be done for the comparison with radiosondes (Figure 14)

**Authors:** We replotted Figure 10 and Figure 14 and rephrased the corresponding interpretations as you suggested. (Page 17, Lines 364-372; Page 19, Figure 10; Page 24, Figure 14)

*Replace "precusion" by "precision"* **Authors:** Done. (Page 20, Line 416)

The sentences "Taking the uncertainty of radiosonde ZWD into account, the improvement of HZWD model over GPT2w model below 2 km seem to be insignificant. However, the validation is based on the same radiosonde ZWD values and the RMS error of ZWD of HZWD is smaller, thus we can reasonably expect that the ZWD of HZWD is closer to true ZWD value than the ZWD of GPT2w in spite of the uncertainty of radiosonde ZWD." are confusing and repetitive. Please rephrase the second sentence and try to split into shorter sentences.

**Authors:** We rewrote the sentence as "Taking the uncertainty of radiosonde ZWD into account, the improvement of HZWD model over GPT2w model below 2 km seem to be insignificant. Nevertheless, we can reasonably think that the ZWD predicted by HZWD is closer to true ZWD due to its smaller RMS error." (Page 21, Lines 421-424)

Mentioning the percentage of improvement for the various layers may be misleading as e.g. 33% of the signal for the top layer is only 2-3 mm, which is insignificant. Authors should put their results into perspective of the magnitude of the ZWD field in each layer.

**Authors:** In the revised paper, we clearly indicated the values of the improvements for each layer (Page 25, Line 494, 498-499). We also added the value of ZWD field in each layer and indicated the significant improvements at the top layer. (Page 25, Lines 500-503) Since the aim of HZWD model is to provide the ZWD corrections for geodetic applications such as wide area augmentation and real-time aircraft positioning, the millimeter level improvements are quite important for the height estimates. We added the interpretation of the effects of the ZWD errors on the height estimates in positioning as well as the corresponding reference. (Page 3, Line 87 to Page 4, Line 88; Page 25, Lines 494-496, Lines 499-500)

**The last paragraph of the paper is an example of the need for language improvement:**

"The HZWD model offers good precision stability in the vertical direction and can meet the requirements of ZWD correction at different heights within the troposphere; however, it can be seen that neither the HZWD, nor the GPT2w, models, i.e., those non- meteorological parameter-based models, performed well in the lower region of the troposphere. In addition, compared with the GPT2w model, HZWD model is a closed model with a limitation to facilitate on-site meteorological observations. Further research is required to assess the variation in and factors influencing of the wet delay and explore the possibility of incorporation of on-site meteorological data." Suggestions:

Remove "," in "GPT2w, models

Replace "lower" by "lowest"

Replace "compared with the GPT2w model, HZWD model is a closed model with" by "compared with GPT2w, HZWD is a closed model with", this way avoiding the repetition of the word "model". Remove "and" in "in and factors"

Authors: We accepted your suggestions and revised the paragraph. (Page 25, Lines 509-514)

**Responses to Editor:**

In the abstract the performance of the HZWD is compared to the performance of the GPT2w and the UNB3m model. If I understand it right, the HZWD requires additional input data that is not required by the other two models. I suggest to mention this in the last sentence of the abstract. You could for example write "however, unlike the GPT2w and the UNB3m model, the HZWD model requires information on the vertical distribution of water vapor." Alternatively, the point that the HZWD model requires additional input data should be mentioned explicitly in the discussion section. In case you strongly disagree on this, I would appreciate if you could briefly comment on this point, perhaps after your revised manuscript has undergone a second round of review (which allows the reviewers to react to your responses).

Authors: The HZWD model developed in this paper is an empirical and climatological model. We derived the model parameters from the decadal time series of ERA-Interim data. As with the UNB3m and the GPT2w models, the HZWD model only needs the time and the position (latitude, longitude, and height) as the input data to calculate the ZWD. Therefore, it indeed does not require the information on the vertical distribution of water vapor when applied to practical conditions. For HZWD model, the most significant difference with the UNB3m and GPT2w models is the division of the height interval with respective functions. For clarity, we added an interpretation as "As with the UNB3m model and GPT2w model, the proposed HZWD model only needs the time and coordinates (latitude, longitude, and height) as input to estimate the ZWD. Compared with the UNB3m and GPT2w models, the unique characteristics of the HZWD model is the height interval divisions with piecewise functions" in Page 13, Lines 255-259.

Also optionally, you could consider whether you want to mention or briefly discuss the potential of numerical weather forecasts to provide this information in the conclusion section.

**Authors:** We added a sentence in the conclusion section as "Additionally, to enhance the model performance in real-time conditions, the potential of the incorporation of numerical weather forecasts data in the HZWD model will be explored in future research." (Page 25, Lines 514-517)